# Towards Informative Few-Shot Prompt with Maximum Information Gain for In-Context Learning

**Hongfu Liu** and **Ye Wang**

School of Computing, National University of Singapore

{hongfu,wangye}@comp.nus.edu.sg

## Abstract

Large Language models (LLMs) possess the capability to engage In-context Learning (ICL) by leveraging a few demonstrations pertaining to a new downstream task as conditions. However, this particular learning paradigm suffers from high instability stemming from substantial variances induced by factors such as the input distribution of selected examples, their ordering, and prompt formats. In this work, we demonstrate that even when all these factors are held constant, the random selection of examples still results in high variance. Consequently, we aim to explore the informative ability of data examples by quantifying the Information Gain (IG) obtained in prediction after observing a given example candidate. Then we propose to sample those with maximum IG. Additionally, we identify the presence of template bias, which can lead to unfair evaluations of IG during the sampling process. To mitigate this bias, we introduce Calibration Before Sampling strategy. The experimental results illustrate that our proposed method can yield an average relative improvement of 14.3% across six classification tasks using three LLMs.

## 1 Introduction

These days the In-context Learning (ICL) ability of pre-trained Large Language Models (LLMs) has garnered significant attention in the community. ICL represents a new paradigm for few-shot learning, which entails performing new downstream tasks based on prompts. These prompts consist of a few input-output pairs, commonly referred to as demonstrations. Such prompts serve as explicit task descriptions for the LLM. LLMs have showcased the formidable capacity of ICL and achieved remarkable performances across various downstream tasks (Brown et al., 2020). In comparison to approaches that involve fine-tuning LLMs on downstream tasks (Devlin et al., 2019; Gao et al., 2021), ICL obviates the need for parameter updates,

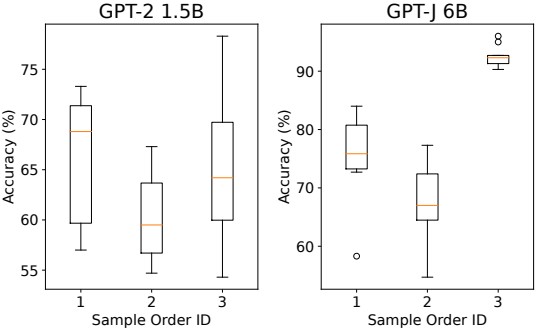

Figure 1: Four-shot ICL performance on SST-2 using GPT-2 XL and GPT-J. Each boxplot summarizes the results of 10 randomly selected prompts for a specific sample order. For a given sample order, the input distribution, demonstration ordering, and prompt formats remain constant. For instance, the sample order [P P N N] denotes the sequence of two positive examples followed by two negative ones.

thereby allowing higher efficiency in adapting to new tasks and easier deployment.

However, ICL tends to suffer from substantial variance in performance. Existing studies attribute it to factors including the input distribution of demonstrations, their ordering, and the prompt formats employed during prompt construction (Zhao et al., 2021; Lu et al., 2022; Zhang et al., 2022; Min et al., 2022). Our investigation reveals that even when the input distribution, the ordering of demonstrations, and prompt formats remain fixed, the random selection of different demonstrations still leads to significant variance, as shown in Figure 1. This observation indicates that data samples within the same category can offer distinct information and contribute differently to the ICL performance. We refer to the ability of a data sample to provide valuable information as its informative ability. To the best of our knowledge, there has been no prior study exploring this aspect in the existing literature.

In this paper, we examine the informative ability

of data examples from the perspective of information theory and investigate its correlation with the ICL performance. Specifically, we assess the contribution of individual data samples to the specific downstream tasks by quantitatively measuring their informative ability. To accomplish this, we propose to evaluate the Information Gain (IG) of prediction, which quantifies the amount of information gained after observing one example candidate in the context. We construct a prompt for each example candidate and utilize the LLM to obtain the corresponding output distribution, thus enabling the evaluation of IG. Furthermore, we uncover the presence of Template Bias, which can lead to biased evaluations of IG. To address this issue, we introduce a Calibration Before Sampling strategy to ensure a fair assessment of IG. Subsequently, we select the example candidates with maximum IG and annotate them as demonstrations for enhancing ICL.

To validate the effectiveness of our method, we conduct empirical evaluations across six classification datasets across three LLMs of varying model sizes. The experimental results demonstrate an average relative improvement of 14.3% on one-shot learning. It is important to emphasize that our proposed method is orthogonal to existing methods such as calibration (Zhao et al., 2021) and re-ordering methods (Lu et al., 2022). Moreover, we demonstrate that our method can be combined with these approaches to achieve further improvements. Additionally, we analyze the relationship between data informative ability with the correctness of target labels and find that data examples with high IG tend to rely more on the accuracy of target labels.

In summary, our contributions can be summarized as follows:

- We investigate the relationship between data informative ability and ICL performance.

- We propose the use of Information Gain (IG) to measure the data informative ability and select demonstrations with maximum IG to enhance ICL performance.

- We identify Template Bias and introduce the Calibration Before Sampling strategy to address it.

- Our proposed method yields significant improvements, achieving an average relative improvement of 14.3% across six classification tasks using three LLMs.

## 2 Related Work

### 2.1 Active Data Sampling for ICL

Active data sampling has been employed in natural language processing tasks since their early stage (Settles, 2009). The primary objective is to achieve comparable or superior performance while reducing the annotation cost. With the advent of pre-trained LLMs, recent studies (Ein-Dor et al., 2020; Yuan et al., 2020; Margatina et al., 2021; Yu et al., 2022) have successfully introduced active learning to minimize the amount of data required for fine-tuning. In the context of the ICL paradigm, the standard ICL involves the random selection of training examples as prompts. However, it has been observed that the performance and stability of ICL can be enhanced by selecting high-quality examples, which aligns with the concept of active data sampling. One common method is to retrieve semantically similar samples for each test query (Rubin et al., 2021; Liu et al., 2021; Hongjin et al., 2023), thereby enabling the utilization of instance-level prompts in downstream tasks. Another approach involves retrieving task-level examples as prompts for all test samples, eliminating the need for instance-level retrieval. (Zhang et al., 2022) introduces reinforcement learning to learn a generalized policy for example selection and (Chang and Jia, 2022) focuses on carefully choosing training subsets to improve stability. However, previous studies either rely on the performances of validation sets as reward signals or require the training of additional models to score example candidates. In contrast, our work centers on retrieving task-level examples from unlabeled datasets for all test samples, without the necessity of extra validation sets or the training of additional models.

### 2.2 Confidence-based ICL Selection

Confidence-based evaluation is widely used for in-context examples. Prior works take the confident model outputs as in-context examples (Wan et al., 2023a) and search confident in-context example organizations in a self-adaptive manner (Wu et al., 2023). In concurrent works, USP (Wan et al., 2023b) utilizes confidence-based prediction for pseudo-demos generation and also handles generation tasks. LENS (Li and Qiu, 2023) proposed informative examples filtering and diversity-guided search method. Our work focuses on addressing the template bias of ICL selection by calibration before sampling.

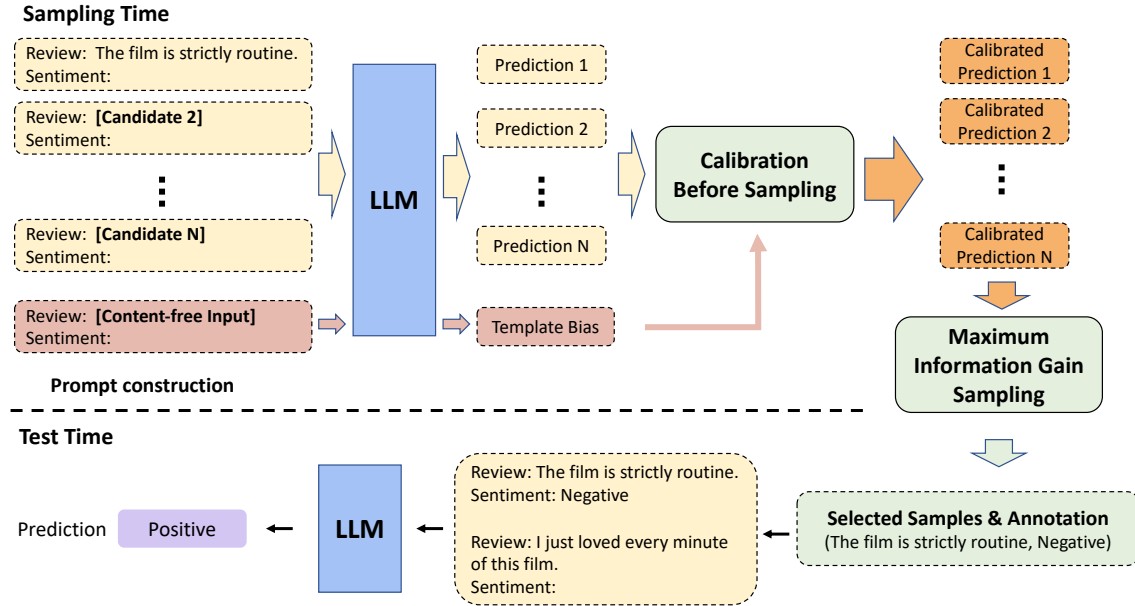

Figure 2: An overview of our proposed method. In Sampling Time, we construct zero-shot prompts, make predictions of example candidates, estimate Template Bias, perform calibration, and sample examples with maximum IG. In Test Time, we annotate selected samples and perform few-shot learning for test samples. **[Candidate N]** denotes the N-th example in the $\mathcal{D}_{unlab}$. **[Content-free Input]** denotes the content-free strings in $\mathcal{D}_{cf}$.

## 3 Methodology

### 3.1 Problem Statement

In this study, we focus on a problem setting that closely resembles true few-shot learning (Perez et al., 2021), which is to retrieve prompts from an unlabeled text dataset, denoted as $\mathcal{D}_{unlab} = \{x_i\}_{i=1}^N$, for a specific task. To accomplish it, we utilize a pre-trained LLM to make predictions on all candidate examples in $\mathcal{D}_{unlab}$, resulting in the corresponding prediction set $\mathcal{Y} = \{\mathbf{y}_i\}_{i=1}^N$, where $\mathbf{y}_i$ represents the normalized predicted label distribution given input $x_i$. The goal is to select a subset $\{x_j\}_{j=1}^K$ from $\mathcal{D}_{unlab}$, where $K \ll N$, in order to facilitate few-shot learning, specifically, $K$-shot learning. We annotate the chosen $K$ examples with their respective target labels $y^t$ and construct task-level prompts using the input-label pairs and task-specific formats (See Appendix A.3 for details). The task-specific prompts are then incorporated as the prefix sequences for test samples.

### 3.2 Information Gain

Information Gain (IG) serves as a metric to quantify the amount of information obtained about a random variable through the observation of another random variable (Ash, 2012). In our context, to measure the informative ability of data examples, we define

the IG as the information obtained in predicted label distribution $Y$ when observing one example candidate $X = x_{ob}$ in $\mathcal{D}_{unlab}$. Specifically,

$$IG(Y, x_{ob}) = H(Y) - H(Y|x_{ob}) \qquad (1)$$

where $H(Y)$ represents the information entropy of $Y$ and $H(Y|x_{ob})$ denotes the conditional entropy of $Y$ given the observation $x_{ob}$. However, computing the exact value of $IG(Y, x_{ob})$ is intractable due to the unknown $H(Y)$. Fortunately, $H(Y)$ remains constant for a given task, allowing us to reframe the problem of sampling examples with maximum IG as selecting those with minimum conditional entropy $H(Y|x_{ob})$. Specifically, considering the LLM parameterized by $\theta$,

$$H(Y|x_{ob}) = -\sum_{y \in Y} p_\theta(y|x_{ob}) \log p_\theta(y|x_{ob}) \quad (2)$$

However, it is challenging to compute $p_\theta(y|x_{ob})$ directly by inputting $x_{ob}$ into the LLM. Instead, we adopt the approach of constructing the zero-shot prompt. One example is shown in the prompt construction of Figure 2. We fill in the task template with text input only and utilize the LLM to make predictions. In other words, each example candidate is taken as a test sample in zero-shot ICL. As such, $p_\theta(y|x_{ob})$ is actually approximated by $p_\theta(y|x_{ob}, T)$, where $T$ denotes the task template.

### 3.3 Template Bias

Taking inspiration from (Zhao et al., 2021), we have observed that the predictive bias also persists even when solely providing the template as input to the LLM. In Figure 3, the red line illustrates the bias associated with the template we employed in SST-2. Notably, when presented with context-free input, the LLM exhibits a tendency for a positive prediction with the possibility over 90%. We refer to this bias as Template Bias [1] (See Appendix A.4 for more details). Essentially, it characterizes the inherent bias present in the zero-shot prompt, whereby the mere utilization of a template prompts the LLM to generate predictions that favor specific answers, in the absence of any demonstration.

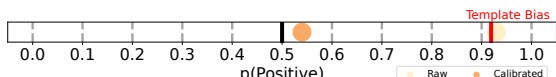

Figure 3: Template Bias on SST-2. The "Raw" point refers to the one before calibration. The "Calibrated" point represents the one after calibration. The balanced line (p=0.5) is bolded.

### 3.4 Calibration Before Sampling

We have observed that the presence of template bias may result in an unfair evaluation of the IG when sampling examples. For instance, the data example located near the red line in Figure 3, exhibits similar information to content-free input but demonstrates high IG (low conditional entropy).

To mitigate this template bias, we propose Calibration Before Sampling strategy[2]. This involves applying the linear transformation[3] (Platt et al., 1999; Guo et al., 2017) to the output probabilities $\mathbf{p} = p_\theta(y|x_{ob}, T)$ to obtain calibrated probabilities $\mathbf{q} = q_\theta(y|x_{ob}, T)$. That is,

$$\mathbf{q} = \sigma(\mathbf{W}\mathbf{p} + \mathbf{b}) \quad (3)$$

where $\sigma$ denotes the normalization function, and the weight matrix $\mathbf{W}$ is constrained to be a diagonal matrix, known as vector scaling. To estimate $\mathbf{W}$,

we leverage the content-free strategy (Zhao et al., 2021). We construct the zero-shot prompt using the task-specific template $T$ and a set of content-free strings $\mathcal{D}_{cf}$, which includes the empty string, "N/A" and "[MASK]". By averaging the output probabilities obtained from each content-free string as input, followed by normalization $\sigma$, we obtain,

$$\mathbf{p_{cf}} = \sigma(\frac{1}{|\mathcal{D}_{cf}|} \sum_{x_{cf} \in \mathcal{D}_{cf}} p_\theta(y|x_{cf}, T)) \quad (4)$$

Consequently, to correct the bias stemming from the template, we set $\mathbf{W} = \mathbf{diag}(\mathbf{p_{cf}})^{-1}$ and $\mathbf{b}$ to be the zero vector. The calibrated conditional entropy can be then computed as,

$$H(Y|x_{ob}) = -\sum_{y \in Y} q_\theta(y|x_{ob}, T) \log q_\theta(y|x_{ob}, T) \quad (5)$$

The example depicted in Figure 3 demonstrates that after calibration, the data sample located around the red line shifts to a position near the balanced line with low IG (high conditional entropy).

The algorithm of our proposed method can be summarized as follows.

---
**Algorithm 1** Maximum Information Gain Sampling with Calibration Before Sampling
---
**Input:**
    unlabeled dataset $\mathcal{D}_{unlab}$, number of examples to be sampled $K$, task-specific template $T$, LLM $\theta$
1: **for** $x$ in $\mathcal{D}_{unlab}$ **do**
2:     Construct prompt for $x$ using $T$
3:     Calculate $p_\theta(y|x, T)$ via LLM
4:     Calculate $\mathbf{p_{cf}}$ using Eq.4
5:     Calculate $\mathbf{q}$ via calibrating $\mathbf{p}$ using Eq.3
6:     Evaluate IG via calculating H(Y|x) using Eq.5
7: **end for**
8: Rank all examples in $\mathcal{D}_{unlab}$ based on IG
**Output:**
    Examples $\{x_j\}_{j=1}^K$ with top $K$ highest IG;

---

## 4 Experimental Setup

**Evaluation Datasets.** We experiment on six text classification datasets including binary sentiment analysis SST-2 (Socher et al., 2013), 6-way question classification TREC (Voorhees and Tice, 2000), 3-way textual entailment CB (De Marneffe

---

[1]We use the term "Template Bias" to distinguish it from the Bias discussed in (Zhao et al., 2021).

[2]We refer to the contextual calibration method in (Zhao et al., 2021) as post-calibration. Calibration alone in our paper refers to the one before sampling proposed in our work.

[3]We apply the transformation to the output probabilities, although it is typically used for logits. The reason is that we only have access to the output probabilities of GPT-3 via OpenAI API. To maintain consistency across different LLMs, we apply the same transformation for GPT-2 XL and GPT-J.

| LM | Method | SST-2 | AGNews | TREC | CB | RTE | DBPedia | Avg |
|---|---|---|---|---|---|---|---|---|
| GPT-2 1.5B | Random | $59.7_{13.2}$ | $39.6_{10.3}$ | $27.1_{5.9}$ | $28.6_{16.1}$ | $53.4_{1.1}$ | $51.2_{14.1}$ | 43.3 |
| | MaxEntropy | $72.2_{10.4}$ | $42.5_{8.1}$ | $25.7_{6.5}$ | $25.0_{19.8}$ | $52.6_{1.4}$ | $30.7_{14.1}$ | 41.5 |
| | MaxIG | $52.8_{0.7}$ | $\mathbf{44.5}_{12.6}$ | $\mathbf{29.8}_{3.7}$ | $39.6_{1.3}$ | $\mathbf{54.4}_{0.6}$ | $\mathbf{56.3}_{10.9}$ | 46.2 |
| | CBS MaxIG | $\mathbf{85.8}_{2.3}$ | $32.9_{12.4}$ | $29.7_{5.0}$ | $39.6_{1.3}$ | $53.9_{1.0}$ | $50.8_{18.5}$ | $\mathbf{48.8}$ |
| GPT-J 6B | Random | $67.5_{6.6}$ | $38.5_{13.8}$ | $38.3_{10.7}$ | $23.2_{7.4}$ | $51.6_{4.1}$ | $61.8_{15.0}$ | 46.8 |
| | MaxEntropy | $75.3_{6.8}$ | $26.9_{4.4}$ | $38.4_{4.5}$ | $26.4_{3.3}$ | $51.0_{4.8}$ | $70.9_{10.8}$ | 48.2 |
| | MaxIG | $65.5_{5.3}$ | $35.5_{11.6}$ | $34.3_{6.1}$ | $31.8_{2.9}$ | $\mathbf{55.3}_{3.6}$ | $64.1_{7.2}$ | 47.8 |
| | CBS MaxIG | $\mathbf{76.8}_{9.1}$ | $\mathbf{39.1}_{12.0}$ | $\mathbf{47.8}_{8.2}$ | $\mathbf{32.5}_{4.4}$ | $\mathbf{55.3}_{3.6}$ | $\mathbf{83.7}_{5.2}$ | $\mathbf{55.9}$ |
| GPT-3 175B | Random | $87.3_{3.7}$ | $62.8_{0.8}$ | $54.8_{1.8}$ | $39.3_{30.4}$ | $56.5_{0.5}$ | $79.7_{5.7}$ | 63.4 |
| | MaxEntropy | $\mathbf{96.5}_{0.5}$ | $63.0_{0.0}$ | $58.7_{2.0}$ | $38.4_{2.7}$ | $\mathbf{62.8}_{0.7}$ | $80.0_{0.7}$ | 66.6 |
| | MaxIG | $92.5_{3.5}$ | $72.3_{2.3}$ | $62.7_{4.0}$ | $\mathbf{41.1}_{0.0}$ | $59.8_{3.8}$ | $81.5_{0.8}$ | 68.3 |
| | CBS MaxIG | $96.2_{0.2}$ | $\mathbf{72.7}_{1.3}$ | $\mathbf{64.3}_{2.0}$ | $\mathbf{41.1}_{0.0}$ | $60.3_{3.3}$ | $\mathbf{87.3}_{0.7}$ | $\mathbf{70.3}$ |

Table 1: Main results for one-shot learning. The last column shows the average accuracies across all tasks. We report the mean and standard deviation across different random seeds. The template of each task is fixed. We bold the best result among all selection methods for each task and each LLM.

.

et al., 2019), RTE (Dagan et al., 2006) from SuperGLUE (Wang et al., 2019), 4-way topic classification AGNews (Zhang et al., 2015), and 14-way DBPedia (Zhang et al., 2015). We use a fixed template (prompt format) for each dataset as per (Zhao et al., 2021). Detailed information regarding each dataset can be found in Appendix A.2.

**Models.** For our experiments, we employ three distinct LLMs with different sizes: GPT-2 XL (1.5B parameters), GPT-J (Wang and Komatsuzaki, 2021) (6B parameters), and GPT-3 davinci (Brown et al., 2020) (175B parameters). We get access to GPT-3 by using OpenAI API.

**Baselines.** In addition to the **Random** baseline, which randomly selects demonstration examples, we also incorporate the widely utilized uncertainty-based baseline **MaxEntropy** in active learning (AL) (Dagan and Engelson, 1995; Settles, 2009). The **MaxEntropy** baseline greedily selects the demonstration example with the highest conditional entropy. The sampling objective of **MaxEntropy** is opposite to that of our proposed method. We refer to our initial IG-based method as **MaxIG**, while the variant that incorporates Calibration Before Sampling (CBS) is denoted as **CBS MaxIG**.

**Other Details.** We use the original ICL, namely the *direct* method, for all experiments in our work. In order to manage the inference cost during sampling, we do not evaluate the entire original train-

ing set. Instead, we first randomly sub-sample $N = 100$ examples from the original training set to form the $\mathcal{D}_{unlab}$ in our experiments. Subsequently, we evaluate all the examples within $\mathcal{D}_{unlab}$ and perform sampling from this subset. For each experiment involving GPT-2 XL and GPT-J, we report the results based on five different random seeds. For experiments involving GPT-3 davinci, we report results using two different random seeds. For evaluation, we sub-sample 300 samples of the test sets for all datasets as per (Zhao et al., 2021; Lu et al., 2022) due to limited resources.

## 5 Results

### 5.1 Main Results

**One-shot Learning.** Our main experiments are conducted in the case of one-shot learning to mitigate potential confounding factors such as the ordering of demonstrations that could influence performance outcomes. By focusing on one-shot learning, we aim to isolate the impact of data informative ability on the performance of ICL. The main results are presented in Table 1. In terms of the average accuracies, our proposed method, CBS MaxIG, exhibits superior performances, achieving relative improvements of 12.7%, 19.4%, 10.9% over the random baseline for GPT-2 XL, GPT-J, and GPT-3 davinci, respectively. These results underscore the effectiveness of our proposed CBS MaxIG. Furthermore, when compared to the MaxIG approach, the Calibration Before Sampling strategy yields improvements in performances for GPT-J and GPT-3

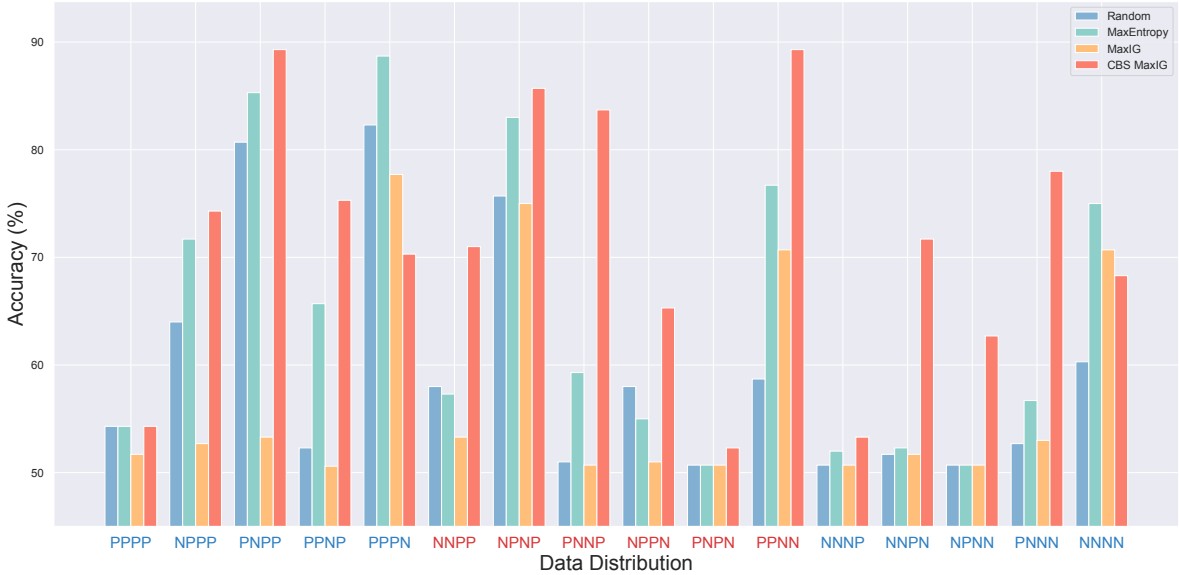

Figure 4: Four-shot performance comparison on SST-2 across different selection methods for different class balances and permutations. [P P N N] denotes two positive examples followed by two negative examples. We display the results for a total of 16 distinct types of four-shot demonstrations. Within this set, 6 types are balanced classes (highlighted in red), while the remaining 10 types are unbalanced classes (highlighted in blue).

davinci, suggesting that this strategy is particularly beneficial for larger models. Notably, our MaxIG-based methods consistently outperform the Random and MaxEntropy baselines, demonstrating the effectiveness of sampling examples with maximum IG and substantiating the validity of employing IG as a metric for evaluating data informative ability.

**Four-shot Learning.** To further evaluate our method for few-shot learning, we extend our experiments to the four-shot learning scenario on SST-2 using GPT-2 XL. We consider all possible combinations of class balances and permutations for the four-shot case[4], which encompass varying label distributions and orders. For each type, data examples are selected using Random, MaxEntropy, MaxIG, and CBS MaxIG methods respectively. This experimental design allows us to examine the impact of data informative ability, given that the class balance and order are fixed for one specific type. The results are presented in Figure 4. We observe that our proposed CBS MaxIG consistently outperforms all baselines in 13 out of 16 cases. For the remaining cases (PPPP, PPPN, and NNNN), we conjecture that factors other than data informative ability, such as the label distribution, may exert a

stronger influence on the performance. We leave comparing the impact of these different factors as future work. Overall, our experimental findings underscore the significance of the data informative ability and demonstrate the efficacy of our method in selecting the most informative data to enhance performance in scenarios involving more than one demonstration.

## 5.2 Integration with Existing Methods

In order to demonstrate that our method is orthogonal to prevailing techniques such as the post-calibration method and the order probing method, and illustrate their collaborative potential with our approach, we conduct experiments on two datasets, namely SST-2 (binary classification) and DBPedia (14-classification), for a comparative analysis.

**Integration with Post-Calibration.** To assess the performance of our method in conjunction with post-calibration, we compare the outcomes of Random and CBS MaxIG approaches on one-shot learning across three LLMs. The results, presented in Table 2, reveal that by employing post-calibration on the selected examples using CBS MaxIG, superior performance is achieved compared to random selection, across different model sizes. Furthermore, it is observed that our method without post-calibration achieves comparable or even superior results to the post-calibration coun-

---

[4]To sample data from different classes, we assume we have access to the target labels of the training set in this experiment. In our initial problem setting, we do not need the target labels during the sampling process.

| | SST-2 | DBPedia |
|---|---|---|
| *GPT-2 1.5B* | | |
| Random (**C**) | $76.7_{1.8}$ | $68.3_{8.7}$ |
| CBS MaxIG | $\mathbf{85.8}_{2.3}$ | $50.8_{18.5}$ |
| CBS MaxIG (**C**) | $74.5_{5.9}$ | $\mathbf{73.5}_{10.5}$ |
| *GPT-J 6B* | | |
| Random (**C**) | $88.6_{1.6}$ | $76.9_{3.4}$ |
| CBS MaxIG | $76.8_{9.1}$ | $83.7_{5.2}$ |
| CBS MaxIG (**C**) | $\mathbf{94.5}_{0.6}$ | $\mathbf{86.7}_{0.8}$ |
| *GPT-3 175B* | | |
| Random (**C**) | $95.8_{0.8}$ | $83.5_{3.5}$ |
| CBS MaxIG | $\mathbf{96.2}_{0.2}$ | $\mathbf{87.3}_{0.7}$ |
| CBS MaxIG (**C**) | $\mathbf{96.2}_{0.2}$ | $86.5_{0.8}$ |

Table 2: One-shot performance using post calibration. **C** denotes the sampling method using post-calibration.

| | SST-2 | DBPedia |
|---|---|---|
| *GPT-2 1.5B* | | |
| MaxEntropy | $\mathbf{72.2}_{10.4}$ | $30.7_{14.1}$ |
| CBS MaxEntropy | $53.1_{1.7}$ | $\mathbf{32.0}_{13.0}$ |
| *GPT-J 6B* | | |
| MaxEntropy | $\mathbf{75.3}_{6.8}$ | $\mathbf{70.9}_{10.8}$ |
| CBS MaxEntropy | $73.5_{4.6}$ | $54.9_{14.2}$ |
| *GPT-3 175B* | | |
| MaxEntropy | $\mathbf{96.5}_{0.5}$ | $\mathbf{80.0}_{0.7}$ |
| CBS MaxEntropy | $88.2_{3.8}$ | $78.2_{3.8}$ |

Table 3: Ablation on Calibration Before Sampling for MaxEntropy in one-shot learning.

terpart specifically for GPT-3 davinci, thereby affirming the effectiveness of our proposed approach.

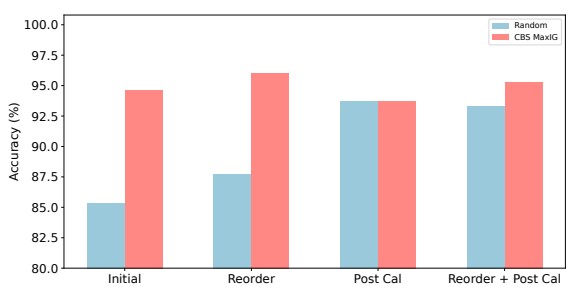

Figure 5: Four-shot performance comparison of four-shot learning on SST-2 using GPT-J. "Initial" represents the original one. "Reorder" denotes the order probing method, and "Post Cal" indicates the post-calibration method. "Reorder + Post Cal" represents the combination of order probing followed by post-calibration.

**Integration with Order Probing.** To assess the performance of Random and CBS MaxIG methods in conjunction with order probing and post-calibration for four-shot learning using GPT-J, we first sample four examples using Random and CBS MaxIG methods[5], respectively. Subsequently, we perform ordering probing and sample the permutation with maximum global entropy on the probing set[6]. The results, depicted in Figure 5, reveal that ordering probing improves the performance for both sampling methods. Furthermore, it is discovered that order probing and post-calibration con-

---

[5]Note that we naively sample the examples with the top-four highest IG without considering label distribution

[6]We utilize the global entropy metric due to its superior performance in the original paper (Lu et al., 2022).

tribute more significantly to enhancing the performance of the Random baseline compared to our CBS MaxIG approach, thereby suggesting that our proposed method is more robust to order and bias factors in comparison to the Random baseline.

### 5.3 Ablation on Calibration Before Sampling

Although the results in Table 1 demonstrate the effectiveness of our Calibration Before Sampling strategy, it should be noted that the evaluation of the MaxEntropy method may also be subject to bias introduced by the template utilized, as the calculation of conditional entropy relies on the output distribution of the LLM. To ensure a fair assessment, we apply the same Calibration Before Sampling strategy to the MaxEntropy method, named CBS MaxEntropy, and report the corresponding one-shot outcomes of SST-2 and DBPedia in Table 3. Notably, a significant decline in performance is observed across all three LLMs, with the exception of DBPedia when employing GPT-2 XL. The performance degradation can be attributed to the fact that examples selected by MaxEntropy may not have the highest entropy. Instead, these examples could correspond to the ones with high IG after calibration. Conversely, examples located near the Template Bias line in Figure 3 are the ones with high entropy after calibration, and CBS MaxEntropy selects those examples. We emphasize this observation as it further reinforces the superiority of our proposed MaxIG-based sampling methods over the MaxEntropy-based approaches. This insight highlights that the MaxEntropy method from conventional active learning, which relies on parameter updates, is not perfectly suitable for ICL where parameters remain static. In such cases, certain examples with high IG contribute more sig-

| Method | Gold | Random | Drop(%) |
|--------|------|--------|---------|
| *SST-2* | | | |
| Random | $67.6_{6.6}$ | $66.7_{8.5}$ | 1.2% |
| CBS MaxIG | $76.8_{9.1}$ | $67.4_{5.3}$ | **12.2%** |
| *DBPedia* | | | |
| Random | $61.8_{15.0}$ | $50.3_{8.1}$ | 18.6% |
| CBS MaxIG | $83.7_{5.2}$ | $36.3_{11.6}$ | **56.6%** |

Table 4: One-shot performance using demonstrations with gold and random labels. The last column shows the percentage of performance drop.

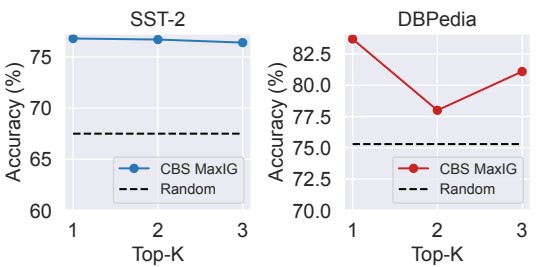

Figure 6: One-shot performance of examples with the Top-K highest IG on SST-2 and DBPedia using GPT-J.

nificantly to ICL compared to uncertain examples with high entropy. The distinction between ICL and traditional active learning settings, and how this distinction influences the choice of sampling strategy, warrants further investigation in future work.

## 6 Analysis

### 6.1 Gold Labels vs. Random Labels

Since our sampling process from the unlabeled training set involves no utilization of target labels, it is pertinent to examine whether the highly informative data derived from it can benefit from the gold labels, as inspired by (Min et al., 2022). In this experiment, we compare the performance of using demonstrations selected by the Random baseline and our CBS MaxIG approach in one-shot learning using GPT-J. For each demonstration, we replace the gold label with a random label from a small discrete set of possible labels, and evaluate the performance accordingly. The results are presented in Table 4.

We observe a substantial decrease in performance when the gold labels are substituted with random labels for demonstrations selected by CBS MaxIG, whereas the drop is comparatively smaller for those randomly selected. This observation suggests that highly informative data heavily rely on the presence of accurate labels. We posit that providing incorrect labels for highly informative data may lead to confusion within the LLM and subsequently result in diminished information gain.

### 6.2 Consistency of Examples with High IG

In order to assess the consistency of performance across other examples with high IG, we individually select the examples with the top-K highest IG values and utilize them as demonstrations in one-shot learning. The results for each selected example

are presented in Figure 6, together with Random baseline depicted with dashed lines. The experimental findings demonstrate that demonstrations with high IG consistently outperform the Random baseline, thereby reinforcing the significance of employing MaxIG-based sampling. Notably, it is observed that the demonstration with the third-highest IG outperforms the one with the second-highest IG. We attribute this discrepancy to potential errors in estimating the IG, which may arise from the utilization of the content-free strategy.

## 7 Conclusion

In this study, we have highlighted the significance of the data informative ability in ICL. We have demonstrated that data samples with varying informative abilities, even when subjected to the same input distribution, order and prompt formats, make distinct contributions to the overall performance of ICL. To address this, we draw inspiration from information theory and proposed to quantify the informative ability by evaluating the information gain of data samples. Moreover, we identify the presence of template bias, which could introduce unfairness in the evaluation of IG. To mitigate this bias, we introduce the Calibration Before Sampling strategy. Through extensive experiments, we have validated the effectiveness of the proposed maximum information gain sampling strategy and calibration before sampling strategy. This validation underscores the reliability of measuring informative ability based on information gain. Furthermore, our experimental findings illustrate that our method is orthogonal to existing approaches and can synergistically collaborate with them to achieve further performance improvements. We hope that our work can provide valuable guidance for the development of data-efficient methods and facilitate the exploration of enhanced data-centric approaches for ICL in the future.

## Limitations

There are several limitations to consider in our work. Firstly, we focus solely on text classification tasks where the information gain can be well-defined based on the prediction distribution and tractable conditional entropy. Future research could extend our experiments to generation tasks. However, this extension poses challenges as it requires a tractable definition of information gain for output distributions that contain open words and have variable lengths.

Additionally, our sampling process does not explicitly consider the diversity of examples. Instead, we prioritize the data informative ability and conduct experiments in one-shot and four-shot scenarios where diversity is not as significant as in other cases with the goal of sampling many samples. Exploring methods to incorporate diversity during the sampling process is of importance for future work.

Another limitation lies in the model-aware evaluation of information gain, which relies on the specific LLM used. This implies that the evaluation results cannot be directly applied to different models. When using a new model, the information gain for each example needs to be recomputed, which incurs additional computational cost. Moreover, the computational cost depends on the size of the training data pool, as each candidate example in the pool needs to be evaluated. Although the parameters of LLMs do not need to be updated, the repeated inferences still consume substantial computational resources, particularly when dealing with extremely large LMs.

## Acknowledgements

We thank anonymous reviewers for their valuable feedback on the paper. We also thank Hengguan Huang and Yisong Miao for their helpful discussions.

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

# A  Appendix

## A.1  Implementation Details

We use Pytorch and Huggingface Transformers in our implementation. We run all our evaluations on a single NVIDIA A40 GPU (48G). Our experiments should be also run on one single GPU of 24G. We access GPT-3 via the OpenAI API[7].

For experiments of GPT-2 XL in Table 1, we rerun the Random baseline due to differences in the training set mentioned in the repository[8]. Nevertheless, our reimplemented results are similar to those reported in (Zhao et al., 2021). Therefore, we report the reimplemented ones for a fair comparison with our proposed method.

## A.2  Dataset Details

We show the statistics of datasets in Table 5. For SST-2 (Socher et al., 2013), AGNews (Zhang et al., 2015), TREC (Voorhees and Tice, 2000), and DBPedia (Zhang et al., 2015), we use their official test sets. For CB (De Marneffe et al., 2019), RTE (Dagan et al., 2006), MNLI (Williams et al., 2018), SNLI (Bowman et al., 2015), and BoolQ (Clark et al., 2019), we use their original validation sets as test sets.

---

[7]https://openai.com/

[8]https://github.com/tonyzhaozh/few-shot-learning

| Dataset | # Classes | # Train | # Eval |
|---------|-----------|---------|--------|
| SST-2   | 2         | 6920    | 1821   |
| AGNews  | 4         | 120k    | 7.6k   |
| TREC    | 6         | 5452    | 500    |
| CB      | 3         | 250     | 56     |
| RTE     | 2         | 2490    | 277    |
| DBPedia | 14        | 420k    | 70k    |
| MNLI    | 3         | 392k    | 9815   |
| SNLI    | 3         | 549k    | 9842   |
| BoolQ   | 2         | 9247    | 3270   |

Table 5: Statistics of evaluation datasets.

## A.3  Template details for different tasks

We show the templates used and corresponding label mappings for different tasks in Table 7.

## A.4  Template Bias for different tasks across three LLMs

We plot the Template Bias of templates used for all tasks across three LLMs in Figure 7 and Figure 8. It is observed that the template bias persists across different tasks and across LLMs with varying model sizes.

## A.5  More experiments

We consider broader NLI and commonsense reasoning tasks. Specifically, we conducted additional one-shot experiments on 3-way classification MNLI, 3-way classification SNLI, and 2-way classification BoolQ using GPT-J 6B. We evaluate them with the same setting in the main experiments. Table 6 shows the performance comparison between our method and baselines. It is observed that ICL without additional reasoning techniques performs poorly on MNLI, SNLI, and BoolQ, which aligns with prior work (Chang and Jia, 2022; Hongjin et al., 2023). Nevertheless, our proposed method still outperforms other baselines on all three tasks.

| Method | MNLI | SNLI | BoolQ |
|--------|------|------|-------|
| Random | $35.9_{7.1}$ | $33.7_{0.0}$ | $58.3_{5.5}$ |
| MaxEntropy | $35.1_{5.2}$ | $37.5_{1.3}$ | $\mathbf{61.1}_{2.7}$ |
| MaxIG | $36.1_{5.3}$ | $37.1_{2.5}$ | $56.6_{4.1}$ |
| CBS MaxIG | $\mathbf{38.3}_{5.5}$ | $\mathbf{40.7}_{3.2}$ | $\mathbf{61.1}_{2.7}$ |

Table 6: One-shot performance on MNLI, SNLI, and BoolQ using GPT-J.

| Dataset | Prompt Template | Label Mapping |
|---|---|---|
| SST-2 | Review: The movie is a desperate miscalculation.
Sentiment: Negative

Review: I hate this movie.
Sentiment: | Positive, Negative |
| AGNews | Article: Toronto Raptors Team Report - November 20,(Sports Network) - The Toronto
Raptors found themselves on the wrong end of a 101-94 decision against the red-hot
Seattle SuperSonics on Friday at the Air Canada Centre.
Answer: Sports

Article: Pioneer replaces plasma TV power supplies, Certain Pioneer TVs have a faulty
power supply. An upgrade is available.
Answer: | World, Sports, Business, Technology |
| TREC | Classify the questions based on whether their answer type is a Number, Location, Person,
Description, Entity, or Abbreviation.

Question: Where is South Bend ?
Answer Type: Location

Question: How many colors are there in the spectrum ?
Answer Type: | Number, Location, Person, Description,
Entity, Abbreviation |
| DBPedia | Classify the documents based on whether they are about a Company, School, Artist, Athlete,
Politician, Transportation, Building, Nature, Village, Animal, Plant, Album, Film, or Book.

Article: Al Gamil is a privately held company based in Djibouti City Djibouti.
Answer: Company

Article: Imperial Botanical Beach Hotel is a hotel in Entebbe Uganda.
Answer: | Company, School, Artist, Athlete, Politician,
Transportation, Building, Nature, Village,
Animal, Plant, Album, Film, Book |
| CB | Richard Breeden hadn't noticed that his new desk had just four telephone lines and one phone.
question: Richard Breeden's new desk had just four telephone lines and one phone. True,
False, or Neither?
answer: True

"I know the one. Yes, it was good though I say it myself." But that doesn't mean I have to
be involved in this kind of nauseous business.
question: she has to be involved in this kind of nauseous business. True, False, or Neither?
answer: | True, False, Neither |
| RTE | IKEA offers fantastic and affordable solutions for your home furnishing needs.
question: Ikea is a home. True or False?
answer: False

I will take a brief vacation with some priest friends after Christmas and then I will go on
retreat at a monastery, Law, reading from a brief statement, told reporters.
question: Law said he plans to take a brief vacation after Christmas and later retreat to a
monastery. True or False?
answer: | True, False |

Table 7: Prompt template and label mapping for different tasks.

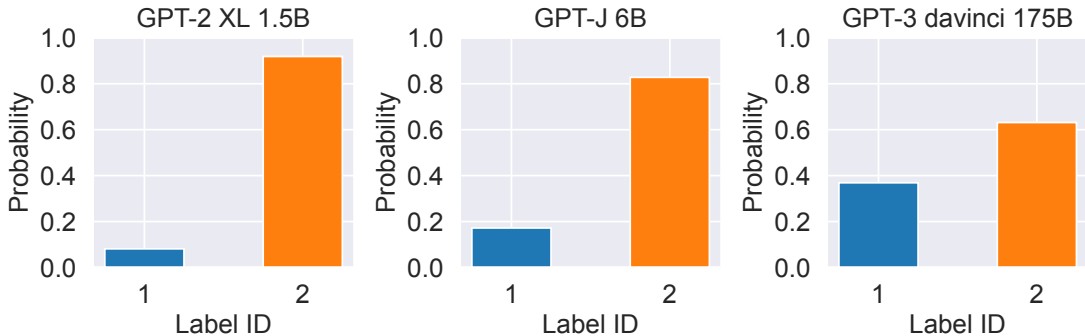

(a) Template Bias of SST-2. Label Dictionary 1: Negative, 2: Positive

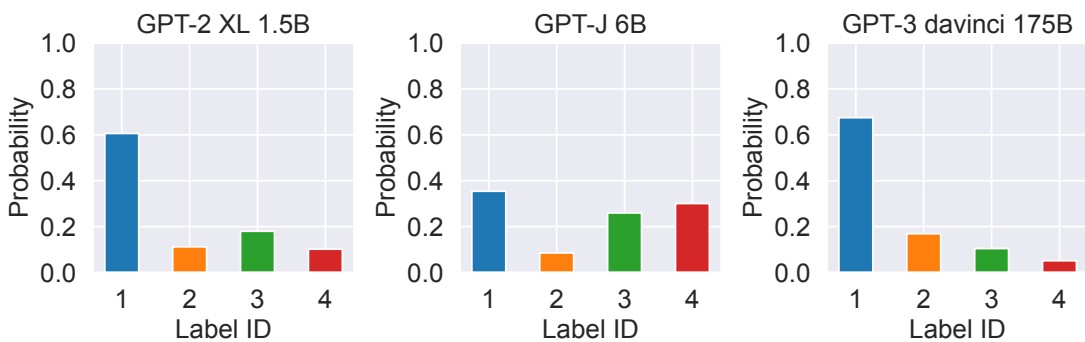

(b) Template Bias of AGNews. Label Dictionary 1: World, 2: Sports, 3: Business, 4:Technology

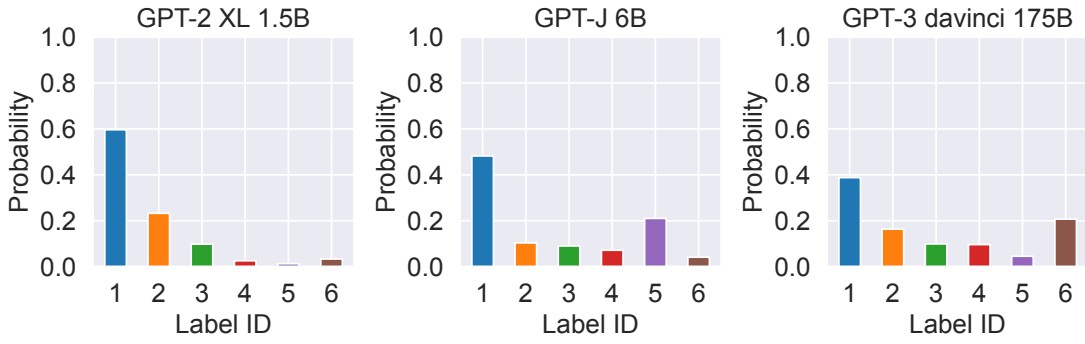

(c) Template Bias of TREC. Label Dictionary 1: Number, 2: Location, 3: Person, 4: Description, 5: Entity, 6: Abbreviation

Figure 7

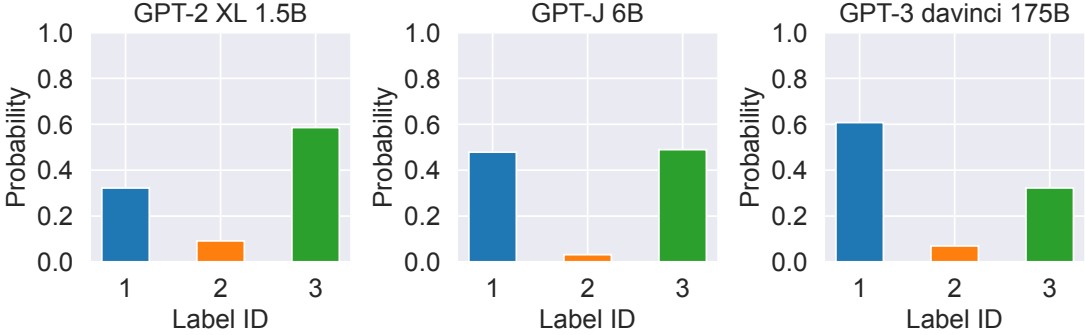

(a) Template Bias of CB. Label Dictionary 1: True, 2: False, 3: Neither

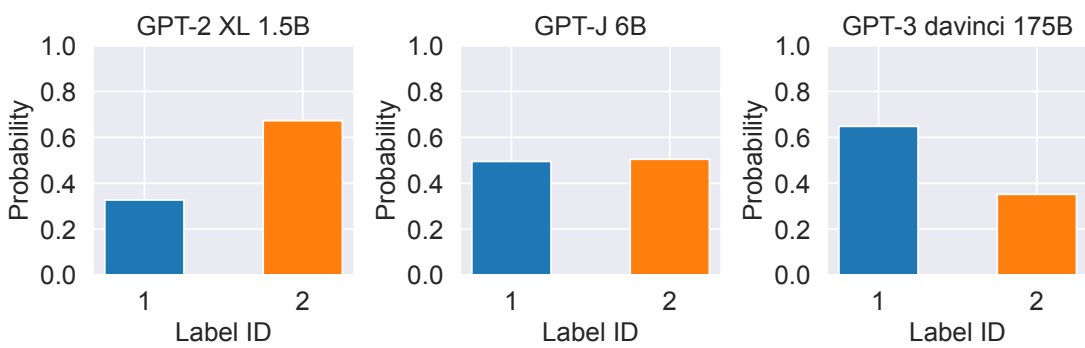

(b) Template Bias of RTE. Label Dictionary 1: True, 2: False

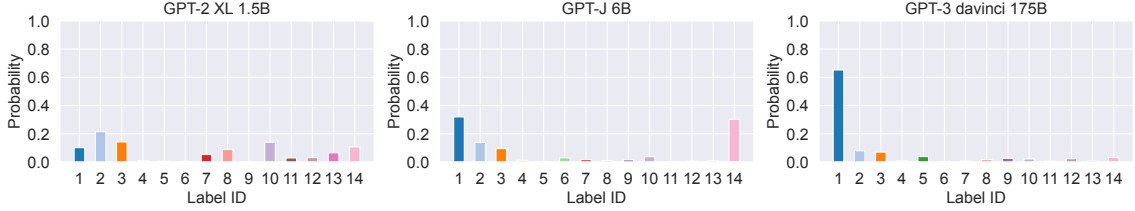

(c) Template Bias of DBPedia. Label Dictionary 1: Company, 2: School, 3: Artist, 4: Athlete, 5: Politician, 6: Transportation, 7: Building, 8: Nature, 9: Village, 10: Animal, 11: Plant, 12: Album, 13: Film, 14: Book

Figure 8