# OpenReview forum: "Towards Informative Few-Shot Prompt with Maximum Information Gain for In-Context Learning"
_EMNLP/2023/Conference — EMNLP 2023 Findings_

### Official Review · Reviewer_CxLY · 2023-07-22

**Soundness:** 4

**Excitement:**

3: Ambivalent: It has merits (e.g., it reports state-of-the-art results, the idea is nice), but there are key weaknesses (e.g., it describes incremental work), and it can significantly benefit from another round of revision. However, I won't object to accepting it if my co-reviewers champion it.

**Missing References:**

[1] Huang, J., Gu, S. S., Hou, L., Wu, Y., Wang, X., Yu, H., & Han, J. (2022). Large language models can self-improve. arXiv preprint arXiv:2210.11610.

[2] Xingchen Wan, Ruoxi Sun, Hanjun Dai, Sercan Arik, and Tomas Pfister. 2023. Better Zero-Shot Reasoning with Self-Adaptive Prompting. In Findings of the Association for Computational Linguistics: ACL 2023, pages 3493–3514, Toronto, Canada. Association for Computational Linguistics.

[3] Wan, X., Sun, R., Nakhost, H., Dai, H., Eisenschlos, J. M., Arik, S. O., & Pfister, T. (2023). Universal Self-adaptive Prompting. arXiv preprint arXiv:2305.14926.

[4] Zhang, Z., Zhang, A., Li, M., & Smola, A. (2023). Automatic chain of thought prompting in large language models. ICLR.

I acknowledge that [2] and [3] are contemporaneous and I have not penalized the authors for not comparing against these works.


**Paper Topic And Main Contributions:**

This paper starts by making the observation that randomly selecting in-context examples leads to high variance in performance, similar to previous observations that shuffling example order leads to performance fluctuation. The authors then propose to select in-context learning examples from unlabeled data with low output entropy (or equivalently high information gain) and use the LLM predictions on these data as in-context examples. The authors further make the observation that the template sometimes induces bias, and they propose calibrating output logits with the CC technique from Zhao et al, 2021 before computing the entropy. They then evaluate several classification tasks on various GPT-style LLMs to show the promise of the proposed method.

**Questions For The Authors:**

Please address my concerns in *Reasons to Reject*.

**Reasons To Accept:**

- ICL has become one of the dominant learning paradigms for NLP, and this paper thus addresses an important problem.
- The methodology design is convincing. Both components of the suggested approach (calibration and ICL selection) are reasonable and intuitive, and the method can be applied in a black-box setting 1) without the need to access model parameters except for the output logits, 2) is capable of working in a zero-shot setup without accessing labels. These are other pluses of the proposed method.
- Experimental validation is largely thorough, and it is nice that the authors have also added several analyses and experiments combining their methods with existing methods to show further enhancements. I have some concerns, though, as listed in the section below.

**Reasons To Reject:**

1. The paper makes three main contributions, namely 1) that ICL is sensitive to random demo selection, 2) ICL selection based on entropy and 3) calibration before sampling. On Point 1, I feel that this is largely expected in light of existing findings that models can be sensitive to even the orders of the examples (which the authors cite) -- I'd tend to think that using *different demos* is a larger source of uncertainty compared to *demos of different order*, so it's unsurprising that randomly sampling demos lead to large variance.


   On 2) and 3), while combining ICL selection and calibration may be new, each of the components is not, and similar ideas have been previously explored. For the first component, the authors are essentially proposing to leverage the low-entropy / high-confidence predictions from the models, and similar ideas have been explored in a number of previous works like [1], where the authors fine-tune the LLMs on the confident outputs based on self-consistency to self-improve. [2] is more directly related, where the authors use confident outputs as in-context examples. [3] is a concurrent work, and the use of USP in the CLS case is almost identical to the proposed method in this paper, but USP also handles additional generative tasks and can work even without accessing the logits in some cases. On the calibration part, the authors essentially lift the CC technique for selection -- while the authors try to distinguish their usage from the one proposed in Zhao et al., 2021, I feel that CC is still essentially used in the same way except that the outputs are now used to compute entropy rather than generating outputs directly.

   Note that I recognize that [2] and [3] are considered contemporaneous to this work, and I have *not* penalized the authors for not discussing them. However, I'd still encourage the authors to discuss the relation with respect to these works when they have a chance to revise their paper.

2. In terms of experiments, I have some suggestions for further improvement:

- In addition to random sampling, there are other suggested approaches for ICL selection, including but not limited to nearest neighbour retrieval (Liu et al., 2021 -- cited in the paper) and diversity based on clustering [4]. I think these are nice to have, even though Liu et al., 2021 and [4] work in slightly different setups ( Liu et al., 2021 assume the availability of golden labels, and [4] is originally proposed for chain-of-thought tasks, like [2], but I think it's easy to adapt the idea of [4] into general tasks) -- note that given the difference in setup, I do not deem the lack of comparison to these works as major weaknesses against the authors.
- It might be beneficial to consider a wider range of tasks outside general classification -- there are ample examples in the GPT-3 paper, for example, where the authors consider broader NLI and commonsense reasoning tasks.

**Reproducibility:**

4: Could mostly reproduce the results, but there may be some variation because of sample variance or minor variations in their interpretation of the protocol or method.

**Reviewer Confidence:**

4: Quite sure. I tried to check the important points carefully. It's unlikely, though conceivable, that I missed something that should affect my ratings.

---

### Official Review · Reviewer_iLKr · 2023-08-06

**Soundness:** 2

**Excitement:**

2: Mediocre: This paper makes marginal contributions (vs non-contemporaneous work), so I would rather not see it in the conference.

**Paper Topic And Main Contributions:**

This study addresses the challenge of demonstration selection in in-context learning. It introduces a novel approach that employs information gain (IG) as a measure to quantify the informativeness of potential examples. The most informative candidates are then selected as demonstration examples within the in-context learning paradigm. Findings indicate that the proposed method typically outperforms baseline approaches in terms of performance.

**Questions For The Authors:**

1. Could you provide some clarification regarding the definition of Y in Sec. 3.2? As per Eq.2, Y appears to represent the prediction distribution for $x_{ob}$, rather than the label distribution in the test dataset. Assuming this is accurate, could you elaborate on how this information gain (i.e., reducing uncertainty on its own prediction given certain LLM) could indicate the informativeness of $x_{ob}$ in enhancing the predictive performance on samples from the testing dataset? If my interpretation is incorrect, could you please explain the correct understanding of this concept?

2. The findings presented in Figure 5 suggest that the proposed method, even without calibration, can yield results comparable to those achieved with calibration. The calibration technique [1] is specifically intended to counteract various biases that may arise during in-context learning in language models. Nevertheless, it seems that no measures were implemented in the proposed method to address these biases. Could you clarify what factors might be contributing to the proposed method's apparent capacity to mitigate these biases?

**Reasons To Accept:**

1. The proposed method boasts cost-efficiency as a key advantage. The selection of demonstration examples is determined solely by the task and template, circumventing the need for selecting new demonstration examples for each incoming test sample. Additionally, there is no need for a large annotated dataset as annotations are required only for the examples once they have been chosen.

2. The idea of using information gain to measure the value of demonstration examples sounds interesting.

**Reasons To Reject:**

1. The proposed method, as described in the methods section, lacks sufficient detail and clarity. See Questions 1 below for detail.

2. This work appears to be missing a comparative analysis with contemporary methods in the field. Even though the authors have considered the MaxEntropy method, there is an absence of referenced literature that could verify its effectiveness in in-context learning scenarios. Additionally, from an active learning perspective, choosing samples with maximum entropy in low-resource conditions (where only a limited number of samples can be selected) may not be the most optimal approach [1]. The absence of comparisons with effective, proven methods makes it difficult to accurately assess the relative efficiency, effectiveness, and innovation of your proposed method.

3. There is a potential issue in the experimental setup that can cause bias in the evaluation. In one-shot in-context learning, the model output can be influenced by the majority label bias [2], where the majority label bias leads the model to predict training answers that frequently appear in the prompt (i.e., the label of the selected one-shot example). The absence of a countermeasure for this bias, especially given that the majority of experiments in this work are conducted under a one-shot learning scenario, raises questions about the accuracy of the results.

[1] Hacohen, G., Dekel, A., & Weinshall, D. (2022). Active learning on a budget: Opposite strategies suit high and low budgets. arXiv preprint arXiv:2202.02794.
[2] Zhao, Z., Wallace, E., Feng, S., Klein, D., & Singh, S. (2021, July). Calibrate before use: Improving few-shot performance of language models. In International Conference on Machine Learning (pp. 12697-12706). PMLR.

**Reproducibility:**

3: Could reproduce the results with some difficulty. The settings of parameters are underspecified or subjectively determined; the training/evaluation data are not widely available.

**Reviewer Confidence:**

4: Quite sure. I tried to check the important points carefully. It's unlikely, though conceivable, that I missed something that should affect my ratings.

---

### Official Review · Reviewer_wbuo · 2023-08-08

**Soundness:** 3

**Excitement:**

4: Strong: This paper deepens the understanding of some phenomenon or lowers the barriers to an existing research direction.

**Paper Topic And Main Contributions:**

This paper examines a few-shot in-context learning (ICL) setting, focusing in particular on the problem of finding samples to be labeled that would be as informative as possible when used as demonstrations to prompt an LLM.
The paper proposes a method to quantify the information gain of a sample. Accordingly, the top K samples will be annotated and used to build the prompt for few-shot ICL. In addition, this paper identifies the phenomenon of "template bias" that consists in a predictive bias due by the template that is used to build the prompt, and proposes a calibration method to mitigate this problem. The combination of methods is then benchmarked on 6 classification tasks with 3 LLMs and compared to a few natural baselines.

### Post-rebuttal change
* Raised Excitement to Strong

**Questions For The Authors:**

- IG is an interesting quantity to optimize. It is then sort of natural to ask, whether it wouldn't be possible to optimize it more directly using prompt tuning techniques. Is this something that the author considered?
- It seems that it should be relatively straight-forward to adapt Algorithm 1 so as to approximately jointly take into account multiple examples in the few-shot learning setting by iteratively greedily growing the prompt one shot at the time. This will come at the cost of running through the unlabeled dataset n_{shots} number of times. Did the authors consider this? This would only make sense if the gain in terms of ICL performance justifies the additional computation.

**Reasons To Accept:**

- Straight-forward but interesting connection between ICL and active learning
- Paper identifies a phenomenon that is termed "template bias" which consists in a systematic shift in the distribution of outputs in few-shot ICL due to the template itself, and propose a calibration procedure to mitigate that problem. This seems like an observation that could have some interesting implications in prompt-engineering more in general.
- The Information Gain method proposed in the paper demonstrates a substantial relative improvement of 14.3% in the six ICL classification tasks where its demonstrated.

**Reasons To Reject:**

- As also clarified in the limitations section, the method is limited to classification tasks, and it is not clear how it could be extended to other settings such as text generation.
- The algorithm proposed in the paper quantifies information gains for each samples independently. Accordingly, in its few-shot incarnation the information gain of each sample is computed in isolation, as opposed to jointly, i.e. taking into account the information gain of entire set of samples. This might leave some performance on the table.

**Reproducibility:**

4: Could mostly reproduce the results, but there may be some variation because of sample variance or minor variations in their interpretation of the protocol or method.

**Reviewer Confidence:**

3: Pretty sure, but there's a chance I missed something. Although I have a good feel for this area in general, I did not carefully check the paper's details, e.g., the math, experimental design, or novelty.

---

### Meta-Review · Area_Chair_oxWe · 2023-09-19

**Recommendation:** 3

**Metareview:**

The authors mostly agree that the submission addresses an important paradigm in NLP, namely in-context learning (ICL), and make an interesting connection between it and active learning, they identify a phenomenon termed template bias which consists of a systematic shift in the distribution of outputs in few-shot ICL due to the template itself, and propose a calibration procedure to mitigate it, the proposed methodology is convincing (calibration and ICL selection), the information gain method demonstrates substantial performance improvements, moreover, it is efficient and forgoes the need for large annotated datasets. However, some weaknesses were also pointed out: the method is restricted to classification tasks, information gains are quantified independently for each sample, however, joint estimation may lead to additional performance gains, the description of the proposed method could use some additional details, the components of the proposed approach are either unsurprising or not new, importantly, rather than penalizing it, the reviewer suggests further discussion, and in the experiments, some baselines are missing (e.g., diversity sampling and clustering) and there is potential for bias in the results. In response, the authors argue that their goal is having a no parameter tuning scheme in favor of data efficiency, that a comprehensive discussion of competing methods will be included in the revision, they acknowledge that max entropy may not be the best strategy for selection and the potential for bias but elaborate on how their information gain framework is likely to counter such bias. Further, the authors agreed to expand the discussion of the proposed methodology in the revision and authors provide additional results for one-shot experiments. Though the reviewers engaged with the authors in the discussion period, one of the reviewers stated that their soundness sore will be increased, however, not reflected in the final score. Nevertheless, such an increase was taken into account.

---

### Decision · Program_Chairs · 2023-10-07

**Decision:**

Accept-Findings

**Comment:**

The authors mostly agree that the submission addresses an important paradigm in NLP, namely in-context learning (ICL), and make an interesting connection between it and active learning, they identify a phenomenon termed template bias which consists of a systematic shift in the distribution of outputs in few-shot ICL due to the template itself, and propose a calibration procedure to mitigate it, the proposed methodology is convincing (calibration and ICL selection), the information gain method demonstrates substantial performance improvements, moreover, it is efficient and forgoes the need for large annotated datasets. However, some weaknesses were also pointed out: the method is restricted to classification tasks, information gains are quantified independently for each sample, however, joint estimation may lead to additional performance gains, the description of the proposed method could use some additional details, the components of the proposed approach are either unsurprising or not new, importantly, rather than penalizing it, the reviewer suggests further discussion, and in the experiments, some baselines are missing (e.g., diversity sampling and clustering) and there is potential for bias in the results. In response, the authors argue that their goal is having a no parameter tuning scheme in favor of data efficiency, that a comprehensive discussion of competing methods will be included in the revision, they acknowledge that max entropy may not be the best strategy for selection and the potential for bias but elaborate on how their information gain framework is likely to counter such bias. Further, the authors agreed to expand the discussion of the proposed methodology in the revision and authors provide additional results for one-shot experiments. Though the reviewers engaged with the authors in the discussion period, one of the reviewers stated that their soundness sore will be increased, however, not reflected in the final score. Nevertheless, such an increase was taken into account.